# Development, Validation and Application of a Bridging ELISA for Detection of Antibodies against GQ1001 in Cynomolgus Monkey Serum

**DOI:** 10.3390/molecules28041684

**Published:** 2023-02-10

**Authors:** Tingting Liu, Yajun Sun, Xiaojie Deng, Lili Shi, Wenyi Chen, Wenjing Fang, Junliang Wu, Xiaotian Fan, Xiaoqiang Chen, Jianhua Sun, Gang Qin, Likun Gong, Qiuping Qin

**Affiliations:** 1Department of Immunoassay and Immunochemistry, Center for Drug Safety Evaluation and Research, Shanghai Institute of Materia Medica, Chinese Academy of Sciences, Shanghai 201203, China; 2GeneQuantum Healthcare (Suzhou) Co., Ltd., Suzhou 215000, China; 3State Key Laboratory of Drug Research, Shanghai Institute of Materia Medica, Chinese Academy of Sciences, Shanghai 201203, China; 4School of Pharmacy, University of Chinese Academy of Sciences, Beijing 100049, China; 5Zhongshan Institute for Drug Discovery, Shanghai Institute of Materia Medica, Chinese Academy of Sciences, Zhongshan 528400, China

**Keywords:** antibody–drug conjugate, immunogenicity, anti-drug antibody, bridging ELISA

## Abstract

Immunogenicity is a major issue associated with the PK, efficacy, and safety evaluation of therapeutic protein products during pre-clinical and clinical studies. A multi-tiered approach consisting of screening, confirmatory, and titration assays has been widely adopted for anti-drug antibody testing. GQ1001, a recombinant humanized anti-human epidermal growth factor receptor 2 monoclonal antibody covalently linked to a cytotoxin of DM1, possesses a novel format of antibody–drug conjugates. In this study, we reported the development, validation, and application of an acid-dissociation bridging enzyme-linked immunosorbent assay for the detection of antibodies against GQ1001 in cynomolgus monkey serum. The sensitivity of the screening assay was 126.141 ng/mL in undiluted serum. The screening assay and confirmatory assay were neither affected by the naïve monkey serum nor by 2% and 5% (*v*/*v*) erythrocyte hemolysates. Moreover, the assay was not subject to interference by 2500 ng/mL of human IgG1 in the samples. Drug interference at low positive control (150 ng/mL) and high positive control (8000 ng/mL) of anti-GQ1001 antibodies was not observed when GQ1001 concentrations were below 3.125 μg/mL and 100 μg/mL, respectively. Furthermore, no hook effect was observed for the positive antibodies in the concentration range of 8 to 64 μg/mL. The validated assay was, thereafter, successfully applied to a single-dose toxicity study of GQ1001. Anti-drug antibody positive rates among dosing animals and testing samples were reported, and no significant impact was found on toxicokinetic outcomes.

## 1. Introduction

Antibody–drug conjugates (ADCs) are biopharmaceutical products that consist of antibodies conjugated with small cytotoxic payloads through linkers, and they have become an efficient type of therapeutics in the field of oncology therapy [1,2,3]. ADCs allow for the targeted delivery of payloads to the tumor site with less impact on normal tissue. Upon reaching the tumor site, ADCs can specifically bind to the antigens and carry the cytotoxins into the tumor cells via endocytosis [4,5]. However, the complexity of the molecular structure of ADCs has raised concerns about their safety. Anti-drug antibodies (ADAs) against ADCs are one of the important factors limiting the application of ADCs [6]. Differences in amino acid sequences between antibodies and endogenous proteins, payloads, linkers, changes in protein structure and modifications, and drug polymerization can all be held accountable for the production of ADAs after administration with ADCs [7,8]. In preclinical studies, ADAs may affect the determination of plasma or serum drug concentrations of ADCs and, thus, impact their pharmacokinetic or toxicokinetic profiles. In clinical practice, ADAs may cause a decrease in or even a loss of efficacy, increased drug toxicity, allergic and hypersensitivity reactions, eventually leading to poor clinical outcomes [6,9]. Therefore, it is suggested that the preclinical ADA results could be used to predict the immunogenicity of drugs in clinical practice. The establishment of assays to detect ADA is crucial in the process of immunogenicity evaluation. Guidance documents have been published by global regulatory agencies to provide regulatory expectations for immunogenicity assessments [10,11].

Human epidermal growth factor receptor 2 (HER2), a receptor for epithelial growth factors (EGF), primarily functions as a transducer of cell growth and proliferation signals. HER2 is expressed on the surface of normal epithelial cells, but it is often highly expressed in tumor cells. The overexpression of HER2 has been proven to be associated with the proliferation, angiogenesis, metastasis and invasion of tumor cells. Once the ligand binds to HER2, it is quickly endocytosed into the cell, thereby terminating the signaling of related pathways. Due to its specific high expression in tumor cells and its endocytic properties, HER2 has been one of the most promising targets for ADC development. DM1, [N2′-deacetyl-N2′-(3-mercapto-1-oxopropyl)-maytansine], is a synthetic derivative of the microtubule-targeted agent maytansine [12,13,14,15,16]. GQ1001 is a novel antibody–drug conjugate developed by GeneQuantum Healthcare for the treatment of HER2-positive solid tumors, consisting of a humanized anti-HER2 antibody and DM1 [17].

To assess the immunogenicity of GQ1001 in a single-dose toxicity study involving cynomolgus monkeys, we developed an acid-dissociation bridging enzyme-linked immunosorbent assay (ELISA) for ADA detection. The developed ELISA method was found to be highly sensitive, specific, and free from matrix effect in the validation study, and it was utilized for immunogenicity assessments in a preclinical toxicity study. In this report, we present the results of the assay validation and the in-study sample analysis in cynomolgus monkeys.

## 2. Results

### 2.1. Method Validation

#### 2.1.1. Screening Cut Point

Screening cut point (SCP) was used to determine whether the sample was screened as ADA-positive or -negative. To establish the SCP, fifty-one cynomolgus serum samples from naïve individuals were analyzed in duplicate by three technicians, in eighteen batches, over four days. SCP was calculated through obtained signal-to-background response ratios (SB). Each serum sample was analyzed over six independent runs. The optical density (OD) value of the pool of normal cynomolgus sera (PNMS) was used as a numerator, the OD value of the negative control (NC) on the same plate was used as a denominator, and the SB was calculated for each single serum sample. Every individual serum sample was analyzed six times, independently. Subsequently, the 25th percentile (Q1) and 75th percentile (Q3) of all 306 effective SB value results were determined with SPSS software (Version 21). Then, the SB values that were less than Q1 − 1.5 × (Q3 − Q1) or greater than Q3 + 1.5 × (Q3 − Q1) were determined as outliers. After the outliers were excluded from the obtained 306 effective SB value results (see Figure 1 and Appendix A), the data were checked for compliance with normal distribution using SPSS software. The results showed that the data did not follow the normal distribution. Thereafter, the data were processed by logarithmic transformation (base 10, lgSB) to determine whether or not they followed the normal distribution. It was found that the data did not follow the normal distribution. Consequently, a frequency analysis of all the unprocessed data after the removal of the outliers was performed, and the 95th percentile, determined as 1.411 (SB), was used as the SCP of the screening assay.

#### 2.1.2. Confirmatory Cut Point

Confirmation cut point (CCP) was used to determine whether the screening-positive sample was confirmed to be ADA-positive or -negative. The result of the confirmation assay was represented as the percent of inhibition ratio (IR%). In order to establish the CCP, fifty-one serum samples of naïve cynomolgus monkeys divided into drug-spiked samples and non-drug spiked samples were analyzed by three technicians, for 18 assay batches, over 4 days. Each serum sample was analyzed over six independent runs (see Figure 2). The Q1 and Q3 of all 305 effective IR% results were processed with SPSS software. Then, the IR% results that were less than Q1 − 1.5 × (Q3 − Q1) or greater than Q3 + 1.5 × (Q3 − Q1) were regarded as outliers. After the outliers were excluded from the obtained 305 effective IR% results, the normal distribution was checked with SPSS software. The results showed that the data did not follow the normal distribution. Consequently, a frequency analysis of all the unprocessed data after the removal of the outliers was performed, and the 99th percentile, determined as 38.2% (IR%), was used as the CCP of the confirmation assay.

#### 2.1.3. Titration Cut Point

The titration cut point (TCP) was used to evaluate the ADA levels for the confirmed ADA-positive samples. The TCP was calculated simultaneously with the SCP and was determined as 99.9th percentile, that is, 1.525 (SB) (see Figure 1).

#### 2.1.4. QC Ranges

As measured and calculated, the SB range of the low positive control (LPC) was 1.411–2.477, and that of the high positive control (HPC) was 24.793–80.548, respectively. The NC range was 0.042–0.100 (OD value).

#### 2.1.5. Sensitivity of the Screening Assay and Precision of the Titration Assay

The sensitivity of the screening assay was investigated by using PNMS to perform two-fold serial dilutions of a high-level testing sample (its concentration was the same as that of HPC). Two technicians conducted the analysis of 12 dilution curves over 3 days and each curve was diluted 12 times. The sensitivity value of each dilution curve was calculated with the following equation:

Sensitivity = FORECAST (SCP, the concentrations above and below the SCP, SB values above and below the SCP)

After this, all 12 of the obtained sensitivity results were logarithmically transformed (base 10), and the Mean and SD of the transformed data were calculated. The sensitivity of the 95% confidence level was calculated using the following formula:

Sensitivity = 10^(Mean + t_0.05, df_ × SD). t_0.05, df_ is the critical value (one-tailed) of the t distribution when α = 0.05 and the degree of freedom is df (df = N−1, where N is the number of samples).

As a consequence, the sensitivity of this method was 126.141 ng/mL in undiluted serum.

The titer of the testing sample was defined as the reciprocal of the maximum dilution when the sample detection value was greater than or equal to the TCP. It was observed that the titers of the 12 dilution curves fell into the range of 640–1280, and the difference between the maximum titer and the minimum titer was not greater than two-fold.

#### 2.1.6. Intra- and Inter-Assay Precision for the Screening Assay and Confirmation Assay

The intra-assay precision of the screening assay and confirmation assay was determined by calculating the CV% of SB values and the CV% of IR% values of six sets of positive control (PC) samples, and the CV% of OD values of NC samples in one batch, respectively.

The inter-assay precision of the screening assay and confirmation assay was investigated by calculating the CV% of SB values and the CV% of IR% values of six sets of PC samples, and the CV% of OD values of NC samples over six batches, respectively (see Table 1).

#### 2.1.7. Effect of Hemolysis and Specificity

Naïve cynomolgus monkey whole blood (K_2_-EDTA as anticoagulant) was stored at −65 °C or lower for at least 12 h to achieve 100% erythrocyte hemolysates. PNMS containing 2% and 5% (*v*/*v*) erythrocyte hemolysates, respectively, was prepared for hemolysis effect evaluation. A high level of anti-GQ1001 antibodies was added to PNMS containing 2% and 5% (*v*/*v*) erythrocyte hemolysates, respectively, to prepare a high-level hemolysis testing sample (HHTS) and a low-level hemolysis testing sample (LHTS). The final concentrations of the testing samples were equal to HPC and LPC, respectively. According to the data shown in Table 2, the ADA analysis (screening assay and confirmation assay) was not affected when the anti-GQ1001 antibodies were added into PNMS containing 2% or 5% erythrocyte hemolysates to produce concentrations of 8000 and 200 ng/mL, respectively.

To assess the specificity of the assay, human IgG1 was used for the validation. The results of the specificity testing are given in Table 2, showing that the ADA assay (namely, both the screening assay and confirmation assay) was not affected by up to 2500 ng/mL of human IgG1 in monkey serum.

#### 2.1.8. Drug Tolerance

The biological samples collected from the dosed animals of a safety study may contain an administered drug that has not been cleared. To test the ability of our established method to detect ADA in the presence of GQ1001, we performed a drug tolerance test. When the concentration of GQ1001 reached a certain concentration (C point), the SB value of the point was greater than or equal to SCP, but the SB value of the next test sample of a higher concentration was less than SCP (the CVs% of the OD values of the two concentration points should be no more than 25.0%). When multiple points satisfied the above acceptance criteria, the first point in the direction of reducing SB values to meet the acceptance criteria was defined as the C point (the CVs% of the OD values of at least 2/3 of the testing samples below the C point should be no more than 25.0%). The results of drug tolerance testing are given in Table 3, showing that the tolerable drug concentration was 100 μg/mL at the positive ADA concentration of 8000 ng/mL, and 3.125 μg/mL at the positive ADA concentration of 150 ng/mL, respectively.

#### 2.1.9. Hook Effect

The hook effect results are given in Table 4, showing that no hook effect was observed in the anti-GQ1001 antibody concentration range of 8–64 μg/mL.

#### 2.1.10. Stability

The results of the stability test are given in Table 5, showing that the anti-GQ1001 antibodies in the testing samples were stable after being stored at room temperature, 2–8 °C for 24 h, −80 °C for 96 days and tolerated 5 F/T cycles.

### 2.2. Method Application

The validated acid-dissociation bridging ELISA was applied to a single-dose toxicity study where the serum GQ1001 concentrations were measured by a PK method that we had previously established. A total of 24 cynomolgus monkeys were randomly assigned into four groups (three/gender/group). The animals were treated with vehicle, GQ1001 (10, 30, or 45 mg/kg), via single intravenous infusion. The blood samples were collected at time points of pre-dose (0 h), and 5 min, 1 h, 4 h, 8 h, 24 h, 48 h, 72 h, 168 h, 336 h, 504 h, 672 h, 840 h, 1008 h post dose. After the intravenous infusion of GQ1001 (10, 30, or 45 mg/kg), serum GQ1001 ADC and TAb concentrations generally peaked rapidly and declined with the time elapse in a roughly biphasic manner, respectively (Figure 3), and GQ1001 exposures (mean C_max_ and AUC_0-t_) increased approximately dose-proportionally (data not shown). No ADA was detected in the animals undergoing the intravenous infusion of vehicle or in the pre-dose samples of the GQ1001. After a single intravenous infusion of GQ1001, the individual positive rates of the two groups (10 or 45 mg/kg) were 16.7% (1/6) and 33.3% (2/6), respectively, and the sample positive rates were 5.6% (1/18) and 11.1% (2/18), respectively. No ADA was detected in the dose group of 30 mg/kg. The total post-dose sample positive rate was 5.6% (3/54), and the total individual positive rate was 16.7% (3/18). In addition, no abnormality was observed on the PK profiles of the ADA-positive animals (see Table 6).

## 3. Discussion

The present assay format incorporates an acid dissociation step into a bridging method to improve the performance of the bridging method for antibody detection. The bridging ELISA is able to detect antibodies regardless of their isotype or the species of origin. However, the detection of antibodies can be difficult in the presence of high levels of antigen in the sample matrix. When excess antigen is present in the blood, it can interfere with the assays through competitive inhibition and/or by forming immune complexes with ADAs [18]. On the other hand, antibodies are able to form much more stable complexes with immobilized antigens compared to those with soluble antigens [19]. Thus, the acid dissociation at a pH of 2 to 3 renders most, if not all, antibodies in a sample free of antigen and is able to re-bind to the antigen upon restoration of the neutral pH. In addition, relatively high concentrations of the biotin-labeled antigen added during the neutralization can compete with the non-labeled antigen for binding to antibodies in the sample [20]. In this way, ADAs in a sample tend to form bridges with biotinylated rather than non-labeled antigen molecules, thereby leading to the more effective detection of ADAs in the presence of antigen.

The assay was developed several years ago on an ELISA platform. The assay sensitivity is sufficient for application in the toxicity study samples, but its drug tolerance is relatively weak, particularly when the positive ADA concentration is at 150 ng/mL. This likely has implications for ADA results obtained from samples where the drug concentrations exceeded the drug tolerance levels, leading to false-negative ADA results. In terms of the toxicity study in which the present assay was applied, the Day 8 and Day 15 samples may be associated with false-negative ADA results. However, the Day 43 samples should be free of the drug interference because the drug concentrations at that time were either below the detection limit of the ADC assay or below the drug tolerance level (3.125 μg/mL) at the positive ADA concentration of 150 ng/mL. Importantly, there is room to further improve the drug tolerance of the assay. This can now be achieved by transferring the assay onto an MSD-based electrochemiluminescence platform. Other technologies, such as antigen capture elution (ACE) [21] and solid-phase extraction with acid dissociation (SPEAD) [22], may also be used to overcome drug interference with the assay, thereby increasing its drug tolerance level.

## 4. Materials and Methods

### 4.1. Materials

GQ1001, monkey anti-GQ1001 polyclonal antibodies, biotinylated GQ1001 (biotin-GQ1001) were provided by GeneQuantum Healthcare Co. Ltd. Streptavidin-HRP A (SA-HRP A) was purchased from R&D Systems (Cat No.: DY998). Tetramethylbenzidine (TMB) was acquired from Aladdin (Cat No.: T117926). Phosphate buffered saline (PBS) was acquired from Thermo Fisher Scientific (Cat No: 10010001). Acetic acid (Cat No: 10000218), tris (hydroxymethyl) aminomethane (Cat No: 30188360), concentrated hydrochloric acid (Cat No: 10011018), sodium carbonate (Cat No: 10019260), sodium bicarbonate (Cat No: 10018960), tween 20 (Cat No: 30189328) and concentrated sulfuric acid (Cat No: 10021618) were purchased from Sinopharm Co Ltd. Bovine serum albumin (BSA), which was used for blocking buffer, was bought from Sangon Biotech (Cat No: A500023).

Naïve cynomolgus monkey sera (NMS) were obtained from the Center for Drug Safety Evaluation and Research (CDSER), Shanghai Institute of Materia Medica (SIMM), Chinese Academy of Sciences (CAS). The pooled naïve cynomolgus monkey serum (PNMS) was prepared from samples of 10 naïve monkeys. The pooled monkey serum was used to prepare quality controls and other test samples. All procedures in this study were in compliance with the animal welfare policies and the guideline of Shanghai Institute of Materia Medica. The protocol was reviewed and approved by the Institutional Animal Care and Use Committee (IACUC) with an IACUC number of 2018-08-RJ-176.

MSX electronic balance (Sartorius, AG, Germany) and microplate reader Infinite F50 (Tecan Trading AG, Zürich, Switzerland), incubator MB100-4A (Allsheng, Hangzhou, China), and microplate washer ELX405R (Biotek, Shoreline, WA, USA) were used.

### 4.2. ADA Samples

The monkey ADA serum samples were obtained from a single-dose toxicokinetic study involving intravenous (IV) infusion. Twenty-four cynomolgus monkeys (3/sex/group, 4 groups in total) were given with vehicle, GQ1001 (10, 30 or 45 mg/kg), respectively, and the serum samples collected at pre-dose (0 h) on Day 1, Day 8, Day 15 and Day 43 were analyzed for ADA. The total drug concentrations in the ADA samples ranged from 200 ng/mL to 300 μg/mL, as determined by a validated sandwich ELISA assay.

### 4.3. Principle of Acid-Dissociation Bridging ELISA

An acid-dissociation bridging ELISA was developed for the detection of anti-GQ1001 antibodies in cynomolgus monkey serum. An acid-dissociation step at pH of 2.2 was performed to render antibodies in a sample free of drug and then ready to re-bind to either free drug or immobilized drug after the neutralization of pH. The bridging part of the method was configured with GQ1001 for capturing ADAs, and biotinylated GQ1001 plus later-added HRP-labeled streptavidin for detecting the bridged ADA-drug complex (see Figure 4). The assay protocol is briefly described as follows: First, microplate wells were pre-coated with GQ1001. Acid dissociation was conducted to reduce possible drug interference in the samples. After the pre-coated microplate was blocked, both biotin-GQ1001 prepared in 1M of Tris-HCl buffer (pH 9.6) and the acid-dissociated samples were added to the microplate. After incubation, the monkey anti-GQ1001 antibodies in the samples were captured as a complex of “immobilized GQ1001-anti-GQ1001 antibodies-biotin-GQ1001”. When SA-HRP A was added to the microplate wells, streptavidin would bind to the biotin-GQ1001 of the complex. Thereafter, TMB was added to the microplate and would react with peroxide in the presence of HRP to form a colorimetric signal (blue). Color development was stopped by the addition of sulfuric acid to the microplate wells, turning the color from blue to yellow. The color intensity (optical density, OD) was measured at 450 nm.

In practice, the method was further divided into three sequential assays: screening assay, confirmation or confirmatory assay, and titration assay. The screening assay was used to identify all samples with signal levels no less than the screening cut point (SCP). The screening assay positive samples were then re-assessed with the confirmation assay for antibodies specific to GQ1001. The confirmation assay positive samples were finally assessed using the titration assay.

Of note, the method was able to detect the antibodies against any parts of the ADC molecule, including the antibody, the bridging part of the antibody and linker, and the toxin. Moreover, the method was also able to detect antibodies against the unconjugated antibodies (mAb).

#### 4.3.1. Screening Assay

For running a screening assay, the microplate was coated with GQ1001 (2 μg/mL) and incubated overnight at 4 °C. After washing with 0.1% PBST thrice, the plate was blocked with 3% BSA. High positive control (HPC, 8000 ng/mL), low positive control (LPC, 200 ng/mL), negative control (NC) and the test samples were initially 20-fold diluted with 0.05 M of citric acid (pH 2.0) and incubated at room temperature (24–27 °C) for 1.5 h with shaking. The acid-treated samples were neutralized by adding 45 μL of 1 M Tris-HCl solution (pH 9.6) containing biotin-GQ1001 (1.6 μg/mL) per 135 μL of acid-treated samples in the coated plate, and incubated for a minimum of 2 h at RT with shaking. Following the incubation, 100 μL of 500-fold diluted SA-HRP A prepared with 3% BSA was added into each well, and the plate was incubated for 1 h with shaking. After washing, TMB substrate was added, and the plate was incubated for 20 min with shaking; the reaction was stopped with 1 M of H_2_SO_4_ before detecting the absorbance signal at 450 nm. Finally, the SB values of the test samples were determined for each individual monkey serum sample.

#### 4.3.2. Confirmatory Assay

The procedure of the confirmatory assay was almost the same as the screening assay, except for the step where individual sample wells were divided into drug-spiked sample wells and drug non-spiked sample wells. Both drug-spiked sample wells and drug non-spiked sample wells were analyzed in duplicate.

Each of the drug-spiked samples contained 135 μL of a testing sample treated with 0.05 M of citric acid and 45 μL of the biotin-GQ1001 solution with 0.6 mg/mL of GQ1001 solution.

Each of the non-drug spiked samples contained 135 μL of a testing sample treated with 0.05 M of citric acid and 45 μL of the biotin-GQ1001 solution without drug solution.

The inhibition rate (IR) was determined for each individual monkey serum sample, as well as the QC sample, in the following way:

IR (%) = [1 − (the OD value of drug spiked sample/the OD value of drug non-spiked sample)] × 100%.

#### 4.3.3. Titration Assay

The confirmation positive samples were initially 2-fold diluted with PNMS and continued until achieving one dilution that gave an SB value lower than the titration assay cut point (TCP). The analytical procedure of the titration assay was the same as that of the screening assay.

#### 4.3.4. Selectivity

At least 10 individual serum samples were used. The monkey anti-GQ1001 antibodies were added to produce the high-level normal selectivity testing sample (HNTS) and low-level normal selectivity testing sample (LNTS). The final concentrations of the testing samples were equal to HPC and LPC, respectively. No positive antibody was spiked into the negative-level normal selectivity testing sample (NNTS). The screening assay and the confirmation assay were performed simultaneously.

#### 4.3.5. Sensitivity and Precision of the Titration Assay

The sensitivity of the screening assay was investigated by using PNMS to perform 2-fold serial dilutions of a high-level testing sample (its concentration was the same as that of HPC). Two technicians conducted the analyses over two days, and 12 curves were analyzed. The sensitivity of each dilution curve was calculated through the SB values and concentrations of the two points above and below the SCP. All the obtained sensitivity results were logarithmically transformed (base 10), and the transformed data were statistically analyzed using t-distribution. The sensitivity of a 95% confidence level was finally obtained by calculation, according to the equation given in Section 2.1.5.

#### 4.3.6. Effect of Hemolysis and Specificity

A high level of anti-GQ1001 antibodies was added to PNMS containing 2% and 5% (*v*/*v*) erythrocyte hemolysates, respectively, to prepare a high-level hemolysis testing sample (HHTS) and a low-level hemolysis testing sample (LHTS). The final concentrations of the testing samples were equivalent to HPC and LPC, respectively. No positive antibody was spiked into the negative-level hemolyzed testing sample (NHTS).

The interference of analogues on the detection of anti-GQ1001 antibodies was investigated by adding human IgG1 (0 to 2500 ng/mL) to a low-level positive testing sample (200 ng/mL) and a high-level positive testing sample (8000 ng/mL), respectively, and thereafter, the resultant samples were analyzed.

#### 4.3.7. Hook Effect

The hook effect involves a signal suppression caused by high concentrations of the ADAs in the sample. To evaluate the hook effect, the monkey anti-GQ1001 antibodies were added to PNMS to produce a series of hook effect testing samples (HETS) (8 to 64 μg/mL), and the screening assay was performed to analyze the testing samples.

#### 4.3.8. Drug Tolerance

The anti-GQ1001 antibodies and GQ1001 were added to the PNMS to produce a series of high-level drug tolerance testing samples (HDTS) (GQ1001: 25 to 300 μg/mL) and low-level drug tolerance testing samples (LDTS) (GQ1001: 0.391 to 12.5 μg/mL), respectively, and the screening assay was performed to analyze the testing samples.

#### 4.3.9. Stability

A high-level stability testing sample and a low-level stability testing sample were prepared by spiking a high level of anti-GQ1001 antibodies into PNMS at the nominal concentrations of 8000 and 200 ng/mL, respectively, and they were then apportioned into aliquots and stored at different conditions (prepared freshly, at RT for 24 h, at 2–8 °C for 24 h, and at −65 °C or lower for 30 days and 90 days).

After at least 24 h of initial storage at −80 °C, the samples were defrosted unassisted at room temperature for 15 min. The samples were transferred back to the original freezer and kept frozen for at least 12 h before the second and third or more defrosting cycles.

### 4.4. Statistics

Data were analyzed using GraphPad Prism 9 (GraphPad Software Inc., San Diego, CA, USA), and the results were presented as the mean and coefficient of variation (CV%) (SD). The data were collected and analyzed by the Magellan Tracker V7.2 (Tecan Trading AG, Switzerland). SPSS Statistics 21 (IBM Corporation, Armonk, NY, USA) was used for the statistical analysis. The concentration–time curve was plotted using Phoenix WinNonlin 6.3 (Pharsight Corporation, Mountain View, CA, USA).

## 5. Conclusions

All the validation results meet the acceptance criteria, which are in line with the related FDA and EMA guidance documents on immunogenicity method validation [10,11], demonstrating the suitability of the use of the acid–dissociation bridging ELISA method for the detection of anti-GQ1001 antibodies in cynomolgus serum. In short, the method with SCP of 1.146 (SB) and CCP of 24.7% (IR%) was sensitive, with a screening assay sensitivity of as low as 126.141 ng/mL in undiluted serum. Furthermore, the method displayed good selectivity and was not affected by the serum matrix and hemolysis. Moreover, the assay exhibited a reasonable level of drug tolerance and was not associated with the hook effect through the concentrations of the positive antibody up to 64 μg/mL. In terms of stability, the positive antibody spiked in the monkey serum tolerated five cycles of freeze–thaw treatment and was stable for 24 h while stored at RT or at 2–8 °C, or at RT after MRD, and was stable for up to 96 days while stored at −80 °C. Finally, while being applied to the samples of a toxicity study performed in cynomolgus monkeys, the method generated reliable results that facilitated the interpretation of the TK assay results, as well as clinical observations, showing that GQ1001 was less immunogenic in the single dose study. Such immunogenicity data are also useful for the design and conduct of future related clinical trials.

## Figures and Tables

**Figure 1 molecules-28-01684-f001:**
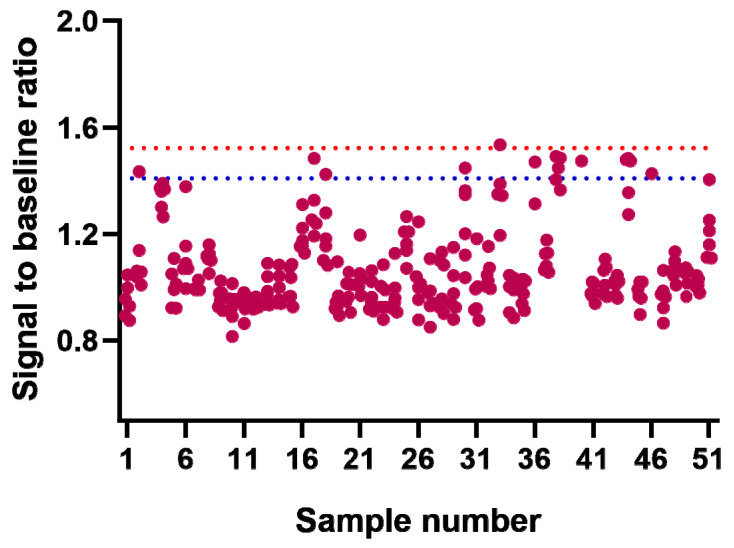
The cut point for the screening assay and titration assay. The screening and titration assays were carried out in six runs by three analysts. The screening cut point (SCP) is indicated with a blue dotted line, and the titration cut point (TCP) is indicated with a red dotted line.

**Figure 2 molecules-28-01684-f002:**
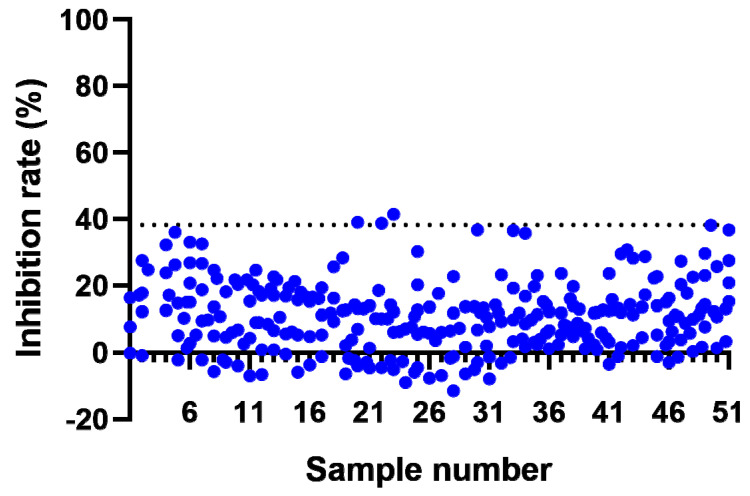
The cut point for the confirmation assay. The confirmation assay was carried out in six runs by three analysts. The confirmatory cut point (CCP) is shown with the blue dotted line.

**Figure 3 molecules-28-01684-f003:**
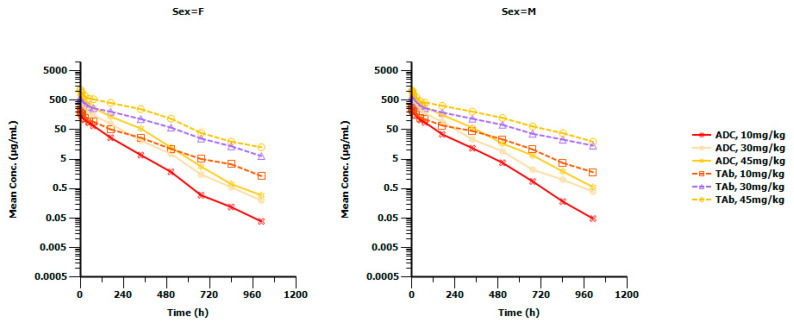
Mean serum concentration–time curves and results after a single intravenous infusion of GQ1001 in cynomolgus monkeys. The figures depict the cumulative concentration of GQ1001 in the serum of cynomolgus monkeys that were randomly assigned into 4 groups (3/gender/group) following single intravenous infusions of 10, 30 and 45 mg/kg GQ1001. Sampling time points include pre-dose (0 h), and 5 min, 1 h, 4 h, 8 h, 24 h, 48 h, 72 h, 168 h, 336 h, 504 h, 672 h, 840 h, 1008 h post dose.

**Figure 4 molecules-28-01684-f004:**
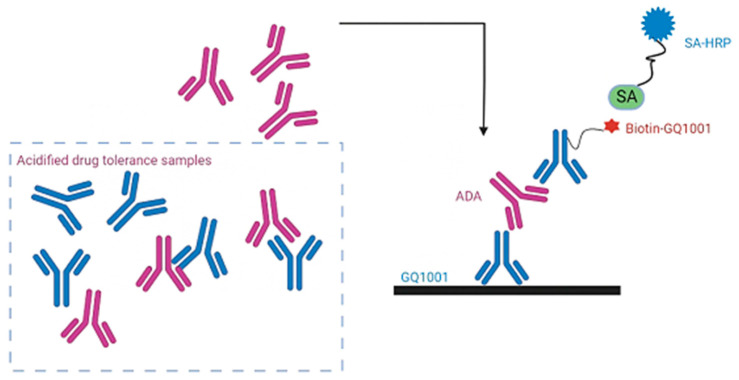
Principle of the acid-dissociation bridging ELISA. An acid-dissociation step at pH of 2.2 is first conducted to render antibodies in a sample free of drug and then ready to re-bind to either free drug or immobilized drug after neutralization of pH. The bridging part of the assay is configured with GQ1001 for capturing ADAs, and biotinylated GQ1001 plus later-added HRP-labeled streptavidin for detecting the bridged ADA-drug complex.

**Table 1 molecules-28-01684-t001:** Intra- and inter-assay precision results.

Test Samples	High-Level Testing Sample (8000 ng/mL)		Low-Level Testing Sample (200 ng/mL)		Negative Control Sample
	SB	IR%	SB	IR%	OD Value
Mean	52.657	98.1	1.957	50.144	0.059
Intra-assay precision (%CV)	2.5	0.1	1.5	2.3	1.4
Inter-assay precision (%CV)	15.5	0.4	10.3	20.3	15.6
Number of replicates	6	6	6	6	6
Number of runs	6	6	6	6	6

**Table 2 molecules-28-01684-t002:** Effect of hemolysis and specificity results.

Test Samples		High-Level Testing Sample (8000 ng/mL)		Low-Level Testing Sample (200 ng/mL)		Negative Control Sample	
		SB	IR%	SB	IR%	SB	IR%
2% Hemolysis		44.736	97.8	1.892	48.1	1.072	6.2
		44.627	97.8	1.887	41.2	1.078	0.1
		41.733	97.5	1.851	44.5	1.127	12.4
5% Hemolysis		41.958	97.3	2.002	39.0	1.222	2.4
		41.902	97.3	2.056	44.1	1.253	9.7
		40.556	97.0	2.060	41.6	1.287	0.3
Human IgG1	2500 ng/mL	38.905	97.7	1.745	47.8		
	500 ng/mL	40.914	97.8	1.715	49.2		
	100 ng/mL	39.212	97.8	1.687	48.5		
	20 ng/mL	37.705	97.6	1.722	49.4		
	0 ng/mL	39.368	97.7	1.782	49.0		

**Table 3 molecules-28-01684-t003:** Drug tolerance results.

Monkey Anti-GQ1001 Antibodies (ng/mL)	GQ1001 (μg/mL)	Mean OD	CV%	SB
8000	300	0.059	0.7	1.068
	200	0.069	0.1	1.247
	100 *	0.098	5.9	1.755 *
	80	0.110	2.0	1.968
	50	0.164	2.3	2.941
	25	0.303	0.6	5.440
	0	2.969	5.0	53.284
150	12.5	0.062	2.4	1.119
	6.25	0.074	1.1	1.322
	3.125 *	0.086	6.7	1.543 *
	1.563	0.091	0.1	1.632
	0.781	0.100	2.7	1.800
	0.391	0.106	6.1	1.900
	0	0.106	2.1	1.906
0	0	0.056	6.3	

Note: * The tolerable drug concentration and the corresponding SB value. SCP: 1.411 (SB).

**Table 4 molecules-28-01684-t004:** Hook effect results.

Monkey Anti-GQ1001 Antibodies (μg/mL)	Mean OD	CV%	SB
64	Overflow	NA	NA
32	Overflow	NA	NA
16	Overflow	NA	NA
8	2.695	2.8	48.379
0	0.056	6.3	

Note: Overflow: the signal overflows; NA: not applicable.

**Table 5 molecules-28-01684-t005:** Stability of anti-GQ1001 antibodies stored at different conditions (including F/T treatment).

Stability	High-Level Stability Sample (8000 ng/mL)			Low-Level Stability Sample (200 ng/mL)		
	Mean OD	%CV	SB	Mean OD	%CV	SB
RT 0 h	2.994	8.6	55.925	0.116	3.4	2.169
RT 24 h	3.198	2.7	56.646	0.121	1.8	2.134
	3.244	0.3	57.455	0.121	3.3	2.150
	3.123	2.4	55.303	0.117	0.2	2.063
2–8 °C 24 h	2.961	4.7	52.444	0.114	0.9	2.023
	3.091	2.2	54.736	0.116	0.1	2.050
	2.959	1.5	52.407	0.115	0.4	2.037
96 days −80 °C	2.453	4.1	49.611	0.092	2.1	1.866
	2.424	4.2	49.030	0.091	0.8	1.841
	2.365	1.8	47.840	0.092	0.5	1.870
5 F/T cycles	3.225	3.7	53.480	0.119	2.0	1.966
	2.961	1.5	49.107	0.116	0.2	1.928
	3.020	2.4	50.079	0.132	23.0	2.196

Note: RT: room temperature; F/T: freezing and thawing.

**Table 6 molecules-28-01684-t006:** Final reported titer results of anti-drug antibody test after intravenous infusions of GQ1001 to cynomolgus monkeys.

Dose Level (mg/kg)	Sex	Time Point			
		Day 1 (Pre-Dose)	Day 8	Day 15	Day 43
0	M	−	−	−	−
	M	−	−	−	−
	M	−	−	−	−
	F	−	−	−	−
	F	−	−	−	−
	F	−	−	−	−
10	M	−	−	−	−
	M	−	−	−	−
	M	−	−	−	80
	F	−	−	−	−
	F	−	−	−	−
	F	−	−	−	−
30	M	−	−	−	−
	M	−	−	−	−
	M	−	−	−	−
	F	−	−	−	−
	F	−	−	−	−
	F	−	−	−	−
45	M	−	−	−	−
	M	−	−	−	−
	M	−	−	−	−
	F	−	−	−	160
	F	−	−	−	−
	F	−	−	−	640

Note: −: negative.

## Data Availability

The data available within the articles.

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
