# Peer review of "Development, Validation and Application of a Bridging ELISA for Detection of Antibodies against GQ1001 in Cynomolgus Monkey Serum"

_molecules, 2023, doi:10.3390/molecules28041684_

Round 1

Reviewer 1 Report

The authors present a careful ADA study of an ADC.

My specific comments are below. Most of these pertain to following the journal's instructions to authors.

Per journal instructions, please provide details on

-the experiments involving monkeys.

-animal welfare

See the Molecules template for more info.

“The animal study protocol was approved by the Institutional Review Board (or Ethics Committee) of NAME OF INSTITUTE (protocol code XXX and date of approval).” for studies involving animals.”

Provide a statement on funding source and conflict of interest.

Delete “It should be mentioned that”

Describe what blinding was used for sample analysis.

What kind of microtiter plates were used? How was CG0001 bound to the plate?

Figure 3. change gender to sex. Monkeys don’t have gender. State the number of animals per group in the figure legend. Which monkeys did not survive?

Table 1. Please include raw data ( i.e. individual value for each measurement) as supplementary data

Fix the author of Ref#10 (should be US FDA) - you have to do it manually

Methods 4.4 statistics states

“…results were presented as mean ± standard deviation”

Please add SD to the tables where appropriate.

Reviewer 2 Report

Antibody-drug conjugates carry a great deal of potential as biopharmaceutical agents, especially as treatments for tumors.  GQ1001 is a recombinant humanized anti-human epidermal growth factor receptor 2 monoclonal antibody covalently linked to a synthetic cytotoxin that is used for the treatment of HER2-positive solid tumors.  However, as with most antibody-drug conjugates, the potential for antibody-drug antibodies to diminish their efficacy is a definite concern.  Whereas detection and quantification of such antibodies is complicated, novel assays to detect and quantify them are sorely needed.  In this highly technical manuscript, a novel acid-dissociation bridging ELISA is developed and validated  for the detection of antibodies against GQ1001 in cynomolgus monkey sera.  

The assay itself is considered highly creative and the authors have done an outstanding job of vetting it for its accuracy, sensitivity and reproducibility.  They have provided stringent confirmation that the accuracy of the assay persists against variations in hemolysis and the Hook effect (see below), as well as the stability of the test results in the face of variations in storage conditions, including temperature and freeze/thawing.  All of these data bode well for this assay to become very effective in monitoring for the development of antibodies that could mitigate the efficacy of GQ1001.  The novelty and demonstrated proficiency of the assay marks it as an important advance in the field.

All of this being said, the manuscript itself is very poorly written and particularly difficult for the reader to navigate.  First, line numbers should be provided.  Also, I found rather annoying the authors’ habit of using abbreviations without defining them when they first appear in the text.  These include the abbreviations DM1 in the abstract, IR%, CV% and PNMS.  One has to look for the definition of these terms later in the text, which is extremely inconvenient in navigating such a complex manuscript.  Also, the authors need to define exactly what the Hook effect is, in order to know the significance of this aspect of the study. 

The assay developed here appears to be very accurate and reproducible.  The authors should present their findings in a manner that renders them more easily accessible for the reader.  As presented here, the manuscript detracts from the significance of the findings.
